# Virus-Encoded Complement Regulators: Current Status

**DOI:** 10.3390/v13020208

**Published:** 2021-01-29

**Authors:** Anwesha Sinha, Anup Kumar Singh, Trupti Satish Kadni, Jayati Mullick, Arvind Sahu

**Affiliations:** 1Complement Biology Laboratory, National Centre for Cell Science, S. P. Pune University Campus, Ganeskhind, Pune 411007, India; anweshasinha28@gmail.com (A.S.); anupkumarrupesh@gmail.com (A.K.S.); truptisatishkadni@outlook.com (T.S.K.); 2Polio Virology Group, Microbial Containment Complex, ICMR-National Institute of Virology, Pune 411021, India; jayati_mullick@hotmail.com

**Keywords:** viral immune evasion, complement, innate immunity, pathogenesis, Poxvirus, Herpesvirus, RCA, CD59, viral RCA

## Abstract

Viruses require a host for replication and survival and hence are subjected to host immunological pressures. The complement system, a crucial first response of the host immune system, is effective in targeting viruses and virus-infected cells, and boosting the antiviral innate and acquired immune responses. Thus, the system imposes a strong selection pressure on viruses. Consequently, viruses have evolved multiple countermeasures against host complement. A major mechanism employed by viruses to subvert the complement system is encoding proteins that target complement. Since viruses have limited genome size, most of these proteins are multifunctional in nature. In this review, we provide up to date information on the structure and complement regulatory functions of various viral proteins.

## 1. Introduction

Viruses face constant challenges from the host due to host resistance, and the environment because of variations in temperature and humidity. Nonetheless, they adapt to these dynamic changes to become the most successful pathogens [1]. The success of virus infection and propagation is determined by many host factors as well as host–virus interactions.

The concept of host resistance to infection was first documented by the Greek historian Thucydides, who provided an eye-witness account of the plague that struck Athens between 430–427 B.C. [2]. However, it was only in the early 20th century that Mechnikov developed the theory of cellular immunity and Ehrlich established the idea of humoral immunity. Yet another critical observation that was made by multiple researchers during the late 19th century was a bactericidal activity of normal serum. Following these studies, in 1895, Bordet showed that bactericidal activity is owing to antibody and a preexisting ‘heat-labile component’ (later dubbed as complement by Ehrlich), which marked the beginning of complement research [3]. He also developed the complement fixation test [4], which became a valuable tool for the diagnosis of various viral diseases and establishing antigenic relationship among viruses. In 1919, he received the Nobel Prize in Medicine for his pioneering work.

The 20th century saw the beginning of the rise of the complement field. It became clear that complement is not a single entity; instead, it is a system that is activated by multiple pathways involving a complex cascade of protease activation. It also became apparent that the system is tightly regulated at various activation steps and lack of regulation results in disease. The characterization of effectors revealed that the system targets pathogens by marking them as ‘non-self’ by opsonins and such tagging of pathogens results in their phagocytosis through complement receptors and lysis owing to loss of their membrane integrity. The complement-activated serum was also shown to have ‘anaphylatoxin’ activity, which was later found to be associated with smooth muscle contraction, enhanced vascular permeability and recruitment of white blood cells [5]. As expected, this activity was demonstrated to contribute to containing localized infections [6]. In the later part of the century, it also became clear that the complement system participates in enhancing pathogen-specific B and T cell responses [7,8] as well as induction of an antiviral state [9].

Thus, it is evident from the above account that complement is a formidable defense against infectious agents, including viruses, as it can not only act directly on pathogens but also enhance innate and adaptive immunity against them. The evolution of viruses, nevertheless, is shaped by constant host-induced pressures [10]. Hence it is not surprising that viruses have developed mechanisms to thwart complement attack. These include encoding molecules to subvert the complement system, acquisition of host complement regulatory proteins, employment of complement receptors for cellular entry and upregulation of host complement regulatory proteins on the infected cells. This review provides an up-to-date account on virus-encoded complement regulators and how they target the complement system.

## 2. Historical Perspective

The first virally encoded molecule that was identified as a complement regulator was herpes simplex virus type 1 (HSV-1) glycoprotein C (gC-1) (Figure 1). In 1982, Harvey Friedman and his group from the University of Pennsylvania observed that human endothelial cells infected with HSV-1 express a C3 receptor [11]. Later, they identified this molecule as gC-1, which has no homology to the known complement receptors [12]. Detailed studies revealed that the molecule is not a receptor; instead, it functions as a C3 regulator [13]. Such mechanism of complement regulation, however, was not unique to HSV-1 as a structurally similar glycoprotein present on HSV-2 could also inhibit complement [14].

In humans, the major complement regulators belong to a gene family termed the regulators of complement activation (RCA). These regulators are exclusively formed by concatenated bead-like domains—the complement control protein (CCP) domains [15]. A homolog of the RCA gene family proteins was first reported in the vaccinia virus in 1988 by Kotwal and Moss [16]. Its biochemical characterization established that the protein essentially functions as human regulators [17]. In a quest to identify such RCA homologs in other viruses, various laboratories examined viral genomes for the presence of sequence homologs. They discovered that apart from poxviruses, RCA homologs also exist in gammaherpesviruses, including Kaposi’s sarcoma-associated herpesvirus [18,19,20]. We now know that functionality in these proteins is dictated by the presence of spatially conserved motifs [21].

The membrane attack complex (MAC) is a critical effector of the complement system that can form pores on the plasma membrane of the target cells. Consequently, its formation on the viral envelope results in loss of viral integrity. In 1992, based on sequence similarity, a group at the Institut fur Klinische und Molekulare Virologie, Erlangen discovered that Herpesvirus saimiri (HVS) encodes a homolog of human CD59 [22]. Later, the laboratories of Stephen Squinto as well as Peter Lachmann, showed that the protein indeed has an ability to block complement-mediated cytolysis [23,24].

Besides RCA and CD59 homologs, various other non-structural viral proteins are known to mediate complement evasion. Among these, non-structural protein 1 (NS1) of flaviviruses is known to subvert the complement system. In the 2000s, the laboratories of John Atkinson and Michael Diamond at the Washington University in St. Louis demonstrated that secreted hexamer of NS1 in particular recruits complement regulators such as factor H and C4b-binding protein onto the infected cells and protect them from complement attack [25,26]. Additionally, the hexamer was also shown to antagonize complement C4 in solution [27]. Ranjit Ray’s laboratory at St. Louis University School of Medicine showed that the non-structural proteins of hepatitis C virus subvert complement by utilizing different mechanisms. NS5A transcriptionally downregulates the expression of complement components like C3, C4, and C9 which participate in complement activation, while NS3/4A proteolytically inactivates C4 [28,29,30].

The complement proteins C1q and mannose-binding lectin (MBL) are pattern recognition molecules which deftly recognize viruses. In the 2010s, it became clear that viruses have also developed mechanisms to block the interaction of these pattern recognition molecules with viruses. The laboratory of Neel Krishna at the Virginia Medical School showed that the astrovirus capsid protein interacts with C1q and inhibits the classical pathway of complement activation likely due to the displacement of protease tetramer C1s-C1r-C1r-C1s [31]. The capsid protein was also shown to interact with MBL and inhibit the lectin pathway [31]. In addition to the capsid protein of astroviruses, the M1 protein of Influenza A virus was also shown to interact with C1q and inhibit the classical complement-mediated neutralization of the virus [32].

Tagging of viruses with C3b is critical for complement-mediated inactivation and clearance. Hence, efficient inactivation of virus tagged C3b is central for viral protection. Recent reports show that RNA viruses like Nipah and Chikungunya display factor I-like activity which can mediate C3b inactivation [33]. This mechanism appears to be unique to viruses as no other pathogen has been shown to exhibit factor I-like activity.

## 3. Viral Complement Regulators

Virus replication is initiated after its attachment to a cell, which is then followed by its cellular entry, uncoating, replication, assembly, and release. Thus, before its entry and after the release, the virus is in the extracellular milieu, while during the other stages, it is inside a cell. An effective innate immune defense, therefore, must target both phases of the virus life cycle. Notably, all the major complement pathways—classical, alternative, and lectin—are potent in targeting cell-free viral particles as well as the virus-infected cells [34,35,36,37]. Moreover, as stated above, the system also boosts other antiviral innate [9] and acquired immune responses [7,8] to limit the viral infection. Studies on the complement subversion mechanisms of viruses have shown that viruses employ structural (i.e., components of the viral particles) as well as non-structural proteins to subvert these responses (Figure 2 and Table 1).

### 3.1. Structural Proteins

#### 3.1.1. Herpes Simplex Virus Glycoprotein C (gC)

HSV is a large (~125 nm) enveloped DNA virus that is classified into two types: HSV-1 and HSV-2. The former causes oral herpes, while the latter causes genital herpes. Although rare, a neonatal herpes infection can also cause a devastating disease.

The glycoprotein C (gC) is one of fifteen envelope proteins of HSV. It mediates the initial interaction of the virus with cells. Along with another envelope glycoprotein termed glycoprotein B (gB), gC binds to the cell surface heparan sulfate proteoglycan and triggers a series of interactions which results in the entry of the virus into the target cell [58,59,60].

Initial work on the glycoprotein C (gC) of HSV-1, termed gC-1, revealed that the protein binds to complement C3b, and is likely a receptor [12]. A structurally similar protein is also present on HSV-2 and is termed glycoprotein C-2 (gC-2). Both gC-1 and gC-2 are homologous in sequence and occupy collinear positions in the genome of the respective viruses [61,62]. gC-1 is a 511 amino acid protein that harbors nine potential sites for N-linked oligosaccharides (N–CHO) [63] and numerous for O-linked oligosaccharides (O–CHO) [64,65]. Importantly, its structure is stabilized by four disulfide bonds [38]. The gC-2, on the other hand, is formed by 480 amino acids having seven potential sites for N-CHOs and several for O–CHOs [62]. Like gC-1, it also contains eight cysteines, but its disulfide arrangement has not yet been determined.

Both gC-1 and gC-2 are distinctly unrelated to any of the known host regulators. Consequently, the mechanisms through which they inhibit complement are also significantly different. Examination of interactions of these viral proteins with complement C3 and its activation products have shown that both bind to native C3 as well as C3b, iC3b and C3c, suggesting their binding site is located in the C3c region of C3 [38]. Both were also shown to provide protection to the virus against complement-mediated neutralization [14,66]. Studies on its mechanism of complement inhibition of gC-1 have shown that it accelerates the decay of the alternative pathway C3 convertase C3b,Bb, but has no effect on its formation [13]. Contrastingly, gC-2 is known to stabilize the C3 convertase [67].

The domain of gC-1 and gC-2 that interact with C3b are conserved in both the glycoproteins [68]. gC-1, however, contains an additional domain located at the amino terminus (residues 33–123) of the protein which accelerates the decay of the alternative complement pathway C3 convertase and prevents interaction of properdin and C5 with C3b [13,38]. The assessment of the importance of the complement-interacting gC-1 domains in the murine model has revealed that the C3-binding domain is much more important than the C5/P-blocking domain and is a significant contributor to the survival of the virus within the host [69]. The affinity of HSV-2 gC for C3b is higher than that of HSV-1 gC, but whether this compensates for the loss of C5/P-interacting domain in gC-2 is not clear [70]. Since gC is vital for virus attachment as well as immune evasion, it was included as a component in the trivalent glycoprotein subunit vaccine to prevent neonatal HSV infections [71].

#### 3.1.2. Astrovirus Capsid Protein (CaPt)

Astroviruses are small (~30 nm) non-enveloped positive-sense single-stranded RNA viruses. They are a leading cause of viral gastroenteritis in infants worldwide [72]. Among its eight serotypes, the serotype 1 is frequently detected.

The human astrovirus capsid protein, which forms the icosahedral capsid shell [73], is crucial for the initial phase of infection. It is synthesized as a 90 kDa protein having three domains—a conserved N-terminal domain, a hypervariable domain, and an acidic C-terminal domain—but requires proteolytic processing for maturation [74,75].

It was noticed that human astrovirus infections do not result in significant inflammation. This led to the idea that the virus may have developed an evasion mechanism against the complement system. To functionally test this, the virus was examined for its ability to inhibit serum complement. The virus showed effective inhibition of serum complement, and for this, it targeted C1. Further examination indicated that the capsid protein of the virus interacts with the A chain of C1q resulting in dissociation of the protease tetramer C1s–C1r–C1r–C1s, which then prevents C1s activation [31]. Because C1q is structurally similar to MBL, a component of the lectin pathway, the capsid protein was also assessed for its ability to bind to MBL. The capsid protein displayed binding and inhibited the lectin pathway activation. It, however, did not show any binding to the MBL mutant K55Q [31], which fails to bind to mannose-binding protein-associated serine protease 2 (MASP-2) [76] suggesting its interaction site on MBL is similar to that of MASP-2. Apart from serotype 1, serotypes 2–4 also inhibited complement activation [44]. It is thus likely that the complement suppressing function of the capsid protein is conserved among the members of human astroviruses.

The capsid protein shares limited sequence homology with human neutrophil defensin-1 (HNP-1), which is known to interact with C1q and MBL, and inhibit the classical and lectin pathways of complement activation [77,78]. The homology resides between the amino acids 79–139 of the capsid protein and the HNP-1. Intriguingly, a 30 amino acid peptide derived from this region displayed binding to C1q and inhibited C1 and C4 activation [79]. The peptide was also tested for its in vivo potency in rats. It exhibited rapid inhibition of complement [80,81].

#### 3.1.3. Influenza Virus Matrix Protein M1

Influenza viruses (~100 nm) are enveloped negative-sense single-stranded RNA viruses. They are of four types: A, B, C and D. Of these, the seasonal flu is caused by type A and B strains and are responsible for approximately 300,000 to 500,000 deaths annually [82].

The matrix protein M1 of influenza viruses is the most abundant viral protein that forms a coat below the lipid envelope. It is a 28 kDa protein with three domains: a N-terminal domain, a middle domain, and a C-terminal domain. The N-terminal and the middle domains are ordered while the C-terminal is flexible and disordered and plays a role in forming supramolecular structures [83]. It plays a key role in virion assembly [84] and inhibits RNA synthesis at the late stage of virus replication [85].

Because M1 is a multifunctional protein, a yeast two-hybrid screen was performed to identify M1-interacting cellular proteins. Interestingly, the study identified C1qA as an M1 binding protein. Direct binding assay confirmed these results and binding analysis with truncation mutants showed that the N-terminal domain of M1 is responsible for this interaction. Examination of interaction of the N-terminal domain with different regions of C1qA showed that the viral protein specifically interacts with the globular region. As expected, the N-terminal domain inhibited the binding of C1qA to IgG as well as the classical pathway-mediated hemolysis. Virus neutralization performed in the presence of the N-terminal domains demonstrated that it is effective in inhibiting the neutralization. In vivo relevance of these results was also determined by performing in vivo studies in a mice model. Administration of the N-terminal domain facilitated influenza virus propagation in the lung and reduced the survival period [32]. What remained unanswered was whether M1 protein is freely available in large enough amounts in the lung to block complement-mediated neutralization of the virus.

#### 3.1.4. Nipah and Chikungunya Virus Proteins with Factor I-Like Activity

Nipah virus (120 to 500 nm) is an enveloped single-stranded negative-sense RNA virus that can cause acute respiratory infection as well as fatal encephalitis in humans. Its recent outbreak in 2018 in the Indian state of Kerala caused 17 deaths. The Chikungunya virus, on the other hand, is a small (60–70 nm) enveloped single-stranded positive-sense RNA virus. It is known to cause a self-remitting febrile viral illness with poly-arthralgia and is known to circulate in Asia, Africa, Europe, and the Americas.

Both these viruses display resistance to complement-mediated neutralization [33,45]. Examination of their surface following incubation with complement showed a marked reduction in C3b deposition, suggesting a lack of complement activation on the viral surface [33,45]. Studies performed to identify the mechanism of C3b inactivation indicated that the viral particles display factor I-like activity. A likely explanation for this could be that these viruses recruit host factor I on their surface, which inactivates C3b with the help of cofactor. However, this premise does not seem to be true because of the following reasons. (i) Nipah virus inactivates only C3b, and not C4b [33], and such inactivation occurs only in the presence of cofactors like factor H and soluble complement receptor-1 (sCR1), but not virus-associated membrane cofactor protein (MCP). (ii) Like that of Nipah virus, Chikungunya virus also inactivated only C3b and not C4b. (iii) The virus associated factor I activity could not be blocked by the mAb (A247) that prevents the function of factor I [45]. The viral proteases that possess factor I-like activity have not yet been identified.

#### 3.1.5. Hepatitis C Virus Core Protein (HCV-CP)

Hepatitis C virus (HCV) is a small (55–65 nm), enveloped, single-stranded positive-sense RNA virus. It causes acute and chronic hepatitis, liver cirrhosis, and hepatocellular carcinoma, and affects about 2% of the human population [86].

The HCV core protein is one of the ten proteins encoded by HCV. It is a multifunctional protein that drives the nucleocapsid formation and also modulates various host cell functions [87]. Structurally it has two distinct domains—N-terminal D1 and C-terminal D2. The D1 domain (120 aa) is hydrophilic and rich in basic amino acids, while the D2 domain (50 aa) is more hydrophobic. Structure-function analysis showed that the D1 domain is involved in RNA binding and oligomerization, whereas the D2 domain is critical for the folding and stability of the core.

Examination of the complement suppressing function of the core protein showed that it regulates complement by two different mechanisms—inhibition of synthesis of complement components and enhancement of expression of complement regulator. In particular, it displays transcriptional repression of the promoters of C3 [47], C4 [28], and C9 [30], though the repression is weak for the C3 promoter. Besides, the core protein was also shown to upregulate the expression of complement regulator decay-accelerating factor (DAF) (CD55) on the hepatocytes [29]. Together, these strategies are expected to protect viral particles as well as the infected cells from complement attack. Additionally, the core protein was also shown to suppress T cell activation following its interaction with complement receptor gC1qR [88], and such suppression was owing to the impairment of IL-2 and IL-2Rα gene transcription [89].

#### 3.1.6. Zika Virus E Protein

Zika virus (ZIKV), an arthropod-borne virus, is a small (~50 nm) enveloped, single-stranded, positive-sense RNA virus. It is one of the prominent members of the family Flaviviridae. Infections caused by this virus are asymptomatic or cause mild symptoms such as fever, rash, red eyes, headache, and joint/muscle pain. However, in some cases, it has also been shown to be associated with neurological and autoimmune complications (i.e., Guillain-Barre syndrome).

The ZIKV E protein (~53 kDa) is one of the three structural proteins of the virus. It is involved in virus entry and hence is a major target for neutralization antibodies. Its ectodomain contains three modules: a central β-barrel module (DI), a finger-like module (DII), and a C-terminal immunoglobulin-like module (DIII). These modules are linked by flexible hinges and the protein is connected to the viral membrane via a helical anchor. On the viral surface, it forms a raft-like structure where 90 E-dimers compact to create the icosahedral symmetry [90].

ZIKV activates the classical pathway due to the binding of IgM antibodies and C1q to the viral surface. Interestingly, though the virus is susceptible to neutralization by high serum concentration (50% serum), it is relatively stable at low serum concentration (10%) [91]. Examination of the mechanism that restricts viral lysis showed that the E protein inhibits the MAC formation. Direct interaction studies showed that E protein binds to C5b-6, C8, and C9, and as a result, blocks polymerization of C9 [48].

### 3.2. Non-Structural Proteins

#### 3.2.1. Hepatitis C Virus NS3/4A Protease Complex and NS5A Protein

HCV encodes six non-structural proteins: NS2, NS3, NS4A, NS4B, NS5A, and NS5B. Among these, the NS3/4A protease complex and NS5A protein have been shown to modulate complement.

Structurally, NS3 protein has a serine protease domain at the N-terminus and a helicase domain at the C-terminus. NS4A, on the other hand, contains an N-terminal membrane-spanning region and a C-terminal cytosolic region. The NS3/4A complex is formed as a result of non-covalent binding of the N-terminal serine protease domain of NS3 to NS4A molecule, where the latter serves as a cofactor. This serine protease complex is required for the proteolytic cleavage of HCV polyprotein precursor. Functional characterization of the protease complex showed that it is necessary for viral replication as well as disruption of the host innate immune system to establish a persistent infection. For example, it is responsible for cleaving host cellular targets involved in the IFN induction pathway.

Its role on the complement system was investigated because, during a proteomic analysis, a fragment of C4 was found to be abundant in HCV carriers. The study specifically examined if NS3/4A protease complex is responsible for the C4 cleavage. In vitro proteolysis assay showed that the protease complex cleaved the γ-chain of C4 and thereby inhibited the classical pathway [46]. In addition, co-transfection of cells with C4 and wild-type NS3/4A, but not a catalytic-site mutant of NS3/4A, produced cleavage of C4γ chain. [46]. Though this explained the higher levels of C4 peptide in HCV carriers, no further efforts were made to determine whether this protease complex is specific for C4 or has the ability to cleave multiple complement components.

The NS5A protein, like that of NS3/4A complex, is also a multifunctional protein. It is necessary for HCV replication, modulation of cellular processes and assembly of infectious virions. It is a zinc-binding phosphoprotein which is rich in proline and is formed by three domains wherein domain I is structured, whereas domains II and III remain unfolded [92]. Domain I has a zinc-binding motif and promotes membrane association, domain II is regulatory and encompasses the interferon sensitivity-determining region, and domain III is necessary for virion assembly [93,94].

Because HCV was shown to inhibit expression of various complement proteins, along with the core protein (discussed above), NS5A was also examined for its ability to transcriptionally downregulate the expression of complement proteins. Initially, NS5A was tested for transcriptional repression of C4. The results showed that it downregulates the expression of C4 by attenuating the expression of interferon regulatory factor 1 (IRF-1), which is significant for IFN-γ-induced C4 expression [28]. Later, NS5A was also investigated for its ability to regulate the C3 synthesis. It repressed C3 expression by inhibiting the expression of IL-1-induced C/EBP- transcription factor [47].

#### 3.2.2. Hepatitis B Virus HBx Protein

Hepatitis B virus (HBV) is a small (42 nm) enveloped double-stranded DNA virus. It is known to cause acute as well as chronic hepatitis and hepatocellular carcinoma (HCC). The current estimate suggests that globally over 2 billion people are infected with hepatitis B infection [95].

The HBV X protein (HBx) is one of the four proteins of the virus. This 17 kDa soluble protein consists of two domains: a negative regulatory domain (1–50 amino acid residues) at the N-terminus and a trans-activating domain (51–154 residues) at the C-terminus [96]. HBx interacts with several cellular proteins, affects cellular processes and participates in the development of HCC [97,98].

Because of HBx’s proposed role in HCC development, a study sought to determine if it is involved in overexpression of CD59, which can provide resistance to infected cells against complement-mediated cytolysis and thereby participate in the tumor cell growth. Initial analysis showed that CD59 is upregulated in clinical HCC specimens. Further study with HBx revealed that it promotes the overexpression of CD59, and this is owing to its binding to the CD59 promoter and down-regulation of let-7i [49]. Thus, the study established a link between HBx and HBV-induced HCC. Following this work, the laboratory further elucidated if HBx upregulates only CD59 or is also capable of inducing the expression of other key complement regulators. They found that the protein also upregulates C4b-binding protein α (C4BPα), which provides protection to the hepatoma cells from complement-mediated cytolysis [50]. The HBx-mediated upregulation of C4BPα was due to activation of transcription factor Sp1 in the promoter of C4BPα [50].

#### 3.2.3. Flavivirus Non-Structural Protein 1 (NS1)

As discussed above, Flaviviruses are small (50 nm) enveloped, single-stranded, positive-sense RNA viruses that are majorly transmitted by arthropods and cause mild febrile illness to severe hemorrhagic manifestations. The prominent members that cause human infections, apart from Zika virus (discussed above), are Dengue virus, Japanese encephalitis virus, Yellow fever virus, West Nile virus, and tick-borne encephalitis virus.

NS1 is the most studied and conserved proteins across flaviviruses. It is also highly conserved among the members. It contains three functionally distinct domains namely a hydrophobic β-roll, an α/β Wing domain and a central β-ladder. The entire structure is stabilized by six disulfide bonds [99,100,101]. Following its synthesis, it undergoes dimerization—intracellularly as well as on the cell surface after posttranslational modification—and accumulates extracellularly as a soluble hexamer [102,103,104,105]. The molecular weight of the dimer ranges from 46–55 kDa owing to differences in glycosylation; each monomer harbors two or three N-glycans. The intracellular dimeric form of NS1 is crucial for efficient viral RNA replication, whereas the soluble hexamer plays a role in immune evasion [106,107,108]. Both cell surface-associated, as well as secreted NS1, are highly immunogenic and have implications in the pathogenesis of flaviviruses.

An early study conducted on the NS1 protein by Brandt et al. [109] showed that it is a potent ‘soluble complement-fixing antigen’. Further exploration revealed that the complement activation was a result of the formation of NS1-antibody complexes. Later, multiple studies showed that its membrane-associated form is capable of recruiting various complement regulators on the infected cells and inhibit classical, alternative and lectin pathway activation. These include factor H, C4b-binding protein, clusterin, and vitronectin [25,26,110,111]. Besides, it was also reported to interact with C9 and inhibit its polymerization and, thereby, the MAC formation [26]. The hexamer form was also shown to inhibit complement but by different mechanisms. A complex of hexamer with C4 and C1s resulted in degradation of C4 leading to consumption of complement [27]. More recently, the soluble form was shown to bind to and protect the dengue virus from lectin pathway-mediated neutralization [112].

#### 3.2.4. Vaccinia Virus Complement Control Protein (VCP) and Smallpox Inhibitor of Complement Enzyme (SPICE)

Vaccinia and variola viruses are two well-known orthopoxviruses. Both are large brick-shaped (~350 by 250 nm) double-stranded DNA viruses. Among these two viruses, variola virus, the causative agent of smallpox, killed about 500 million people. Vaccinia virus, on the other hand, does not cause disease in immunocompetent individuals but is known to cause repeated outbreaks in dairy cattle in India and Brazil [113,114]. Notably, vaccinia virus was used as a vaccine to eradicate smallpox.

Vaccinia virus (VACV) encodes a homolog of human RCA proteins dubbed vaccinia virus complement control protein (VCP). It is one of the 20 immunoregulatory proteins of VACV. Incidentally, it is also the major protein secreted by the virus-infected cells [16]. The characteristic feature of the RCA family proteins is that they are entirely formed by bead-like domains termed complement control protein (CCP) domains, and these are linked by short amino acid linkers. Further, the structure of the CCP domain is stabilized by two disulfide bonds. VCP is formed by four such CCP domains, which are linked by 4-amino acid linkers. It also contains an unpaired cysteine at the N-terminus that helps in forming function enhancing disulfide-linked homodimers [56].

Original functional characterization of VCP was performed following its purification from the culture supernatant of VACV-infected cells. The protein effectively blocked the classical pathway hemolytic activity [17]. Later, it became clear that like human RCA proteins, VCP supports the inactivation of complement proteins C3b and C4b and decays C3 convertases, the central enzymes of the complement system [53,54,115]. Studies using truncation domains identified that a minimum of three CCP domains (CCP1-3) are required for all of its functional activities, but the presence of the fourth domain (CCP4) was crucial for the optimum activity [116]. Next, to dissect the role of its individual domains, a domain swapping mutagenesis was performed which mapped CCP2-3 as critical domains for its ability to support the inactivation of C3b and C4b, and CCP1 as essential for its ability to decay C3 convertases [117]

The importance of VCP in VACV virulence was addressed using rabbit, guinea pig and mice intradermal models. Together, these studies established that VCP contributes to virulence by inhibiting complement, which then results in decreased antibody and T cell responses [118,119]. The significance of its regulatory activities in VACV virulence was substantiated using a panel of function-blocking mAbs against VCP in a rabbit intradermal model. The study suggested that principally its ability to support the inactivation of C3b and C4b contributes to virulence [120]. Understanding the effectiveness of VCP as a complement inhibitor in inflammatory conditions is underway by a few groups [121,122].

The complement inhibitor encoded by variola virus was named smallpox inhibitor of complement enzymes (SPICE) [51]. Because variola virus displayed strict human tropism, Rosengard laboratory hypothesized that SPICE must be better suited than VCP to overcome human complement, though they differ only in 11 amino acids. This supposition proved correct, and SPICE was found 100-fold more potent in inactivation of human C3b and 6-fold more efficient in inactivation of human C4b, in comparison to VCP [51]. Later, Sfyroera et al. [123] showed SPICE to be 75-fold and 1000-fold more potent than VCP in inhibiting the human classical and alternative pathways, respectively. A comparison of the C3 convertase decay activity of SPICE and VCP by Liszewski et al. [56] demonstrated SPICE to be 10-fold more active in decaying the classical pathway C3 and C5 convertases, respectively. The same group determined that SPICE attaches to cells via its glycosaminoglycans to efficiently regulate complement on the cell surface [124].

The first effort to understand the functional advantage of SPICE over VCP was attempted using an electrostatic modelling approach. The study showed that E108K and E120K substitution in VCP significantly augmented its C3b cofactor activity [123]. A systematic analysis on the involvement of each of the 11 residue variants in SPICE using point mutants, and tetra- and penta-mutants demonstrated that four out of eleven residues (Y98, Y103, K108 and K120), residing in CCP2, are sufficient for the enhanced C3b and C4b cofactor activity of SPICE over VCP [52]. Studies on the species specificity of SPICE and VCP showed that both exhibit strong selectivity, which is consistent with the species tropism of variola and vaccinia viruses, i.e., SPICE is human complement specific and VCP bovine complement specific. Remarkably, the determinant for the switch in the selectivity of SPICE and VCP is the presence of oppositely charged residues in CCP domains 2–3 [125,126].

#### 3.2.5. Cowpox Virus Inflammation Modulatory Protein (IMP)

Cowpox virus (CPV) is a zoonotic virus which is closely related to vaccinia and variola viruses. It exhibits a broad host range [127] and is known to cause human infections limited to Europe and adjacent regions of Russia [128]. The virus also has historical importance, as it was used by Edward Jenner to immunize against smallpox.

The genome analysis of CPV showed that like that of vaccinia and variola, it also encodes an RCA homolog [129]. Like VCP and SPICE, it also has four CCP domains, which are separated by 4 amino acid linkers. Examination of its activity against complement showed that it is indeed capable of inhibiting complement-mediated hemolysis [55]. The protein was named the inflammation modulatory protein (IMP) because it suppressed inflammation. Specifically, when CPV lacking IMP was injected into mouse footpad, it produced much more swelling and hemorrhage compared to that produced by the wild-type CPV [55]. No efforts, however, were made to further characterize IMP with respect to its complement regulatory activities.

#### 3.2.6. Monkeypox Inhibitor of Complement Enzyme (MOPICE)

Like other orthopoxviruses, monkeypox virus is also a brick-shaped (~200 by 250 nm) double-stranded DNA virus. It is important to mention here that currently, it is recognized as the most critical reemerging zoonotic orthopoxvirus infection in humans. Though the virus is restricted to West and Central Africa’s rainforest regions, its outbreak occurred in the United States in 2003 [130].

The monkeypox virus complement regulator monkeypox inhibitor of complement enzyme (MOPICE) is a 24 kDa soluble protein secreted by the virus-infected cells. It is an ortholog of VCP with a single nucleotide deletion causing a frameshift mutation in CCP4. As a result, it contains only three CCP domains [56]. Analysis of the genomic sequences of the more virulent Congo basin strain of monkeypox virus with the less virulent strain of West Africa revealed that MOPICE is present only in the virulent strain [131].

MOPICE targets complement proteins C3b as well as C4b [56]. Comparative binding of human C3b to MOPICE, VCP and SPICE showed that despite truncation in the fourth CCP in MOPICE, it possesses the ability to bind human C3b more proficiently than VCP; its binding was lower compared to SPICE. A similar comparison of the binding to human C4b showed that its binding was comparable to VCP, but less compared to SPICE. Examination of its ability to inactivate C3b (cofactor activity) showed that it is as potent as VCP, but less compared to SPICE. Concerning inactivation of C4b, it is more potent than VCP but less than SPICE. Unlike other viral RCA regulators, MOPICE lacks the decay acceleration activity (DAA) for C3 and C5 convertases of classical and alternative complement pathways, which is likely due to the lack of CCP4 [56].

Because MOPICE is present only in the virulent form of monkeypox virus, it was hypothesized that it might play a significant role in the monkeypox pathogenesis. Thus, to validate this assumption, in vivo infection studies were undertaken by two groups [132,133]. Infection studies in black-tailed Prairie dog challenge model using recombinant viruses—West African strain inserted with MOPICE and Congo basin strain deleted of MOPICE—showed that insertion of MOPICE led to a minor change in disease manifestation, but the removal of MOPICE delayed the signs of the disease [133]. The study concluded that the protein plays a moderate role in disease pathogenesis. Infection studies in the rhesus macaques, on the other hand, showed that deletion of MOPICE results in enhanced viral replication and subdued immune response against the virus [132].

#### 3.2.7. Ectromelia Virus Inhibitor of Complement Enzymes (EMICE)

Ectromelia virus is another Orthopoxvirus which causes the disease mousepox. The virus was originally isolated in the 1930s from the laboratory strain of mouse and found to easily transmit between the wild and the laboratory mice [134]. It is very closely related to smallpox and monkeypox viruses, with similar pock type rash though the infection route is through skin abrasion, unlike variola virus which occurs via inhalation.

Structurally, ectromelia virus inhibitor of complement enzymes (EMICE) shares 90% homology with the other poxviral RCAs with a major difference in CCP1 [57]. Despite this difference, EMICE possesses C3b and C4b binding activity comparable to MOPICE. The EMICE protein is more similar to VCP having only 18 amino acid differences with the deletion of 2 amino acids. Infection experiment with ectromelia virus in murine cells showed that the protein is secreted from the infected cells within 4 to 6 days post-infection, i.e., much before releasing intracellular mature virions [57]. Examination of complement regulatory activity of recombinant EMICE demonstrated that it is effective in protecting virus-infected cells as well as mature virions from mouse complement [57]. Mechanistic studies pointed out that the protein supports inactivation of C3b as well as C4b, and is effective in the dissociation of the classical pathway C3 convertase [57].

#### 3.2.8. Herpesvirus Saimiri Virus Complement Control Protein Homolog (HVS CCPH)

Herpesvirus saimiri (HVS), a gammaherpesvirus, is a T-lymphotropic tumor virus found in its natural host squirrel monkey. It is a large (170 nm) double-stranded DNA virus that promotes acute leukemia and T cell lymphoma in several species of New World primates such as owl monkeys, common marmosets, and cottontop tamarins [135,136]. It is divided into three subgroups (A, B, and C) based on its oncogenic potential. The subgroup C is capable of transforming simian and human T-lymphocytes to permanently growing T-cell lines in vitro [137,138].

HVS encodes two homologs of the complement regulatory proteins that inactivate complement pathways at distinct steps: (i) a homolog of RCA encoded by ORF4, termed complement control protein homologue (CCPH) [22] and (ii) a homolog of the terminal complement inhibitor CD59 (HVSCD59) encoded by ORF15 [139,140] (described below). Due to differential splicing, the HVS CCPH gene codes for two forms of the protein—a full-length membrane form (mCCPH) comprising four extracellular CCP domains and a transmembrane domain, and a secretory form (sCCPH) that contains only four CCP domains. The CCP1 and CCP3 are predicted to have N-glycans.

Studies by Fodor et al. [39] on the initial characterization of mCCPH illustrated its capacity to block C3b deposition and complement-mediated lysis when expressed on cells. Subsequently, the mechanism of inactivation of complement by CCPH was studied by expressing sCCPH [40]. It revealed that sCCPH binds to both C3b and C4b and inactivates these molecules with the help of factor I. The same study identified Arg118 (which corresponds to K120 of SPICE) as a critical residue that assists in C3b inactivation. The protein was also shown to possess potent decay activity against the C3 convertase of the classical pathway, and to a limited extent, against the alternative pathway C3 convertase [40]. Domain-wise analysis of the CCPH suggested that though all the four domains are required for its full activity, CCP2 alone can function as a complement regulator [141]. The protein was also subjected to extensive mutagenesis, which identified the involvement of specific positively charged and hydrophobic residues in its regulatory activities [142].

#### 3.2.9. Kaposi’s Sarcoma-Associated Herpesvirus Inhibitor of Complement Activation (Kaposica)

Kaposi’s sarcoma-associated herpesvirus (KSHV) or the human herpesvirus 8 (HHV-8) is etiologically linked to Kaposi’s sarcoma. Additionally, it is also linked with pleural effusion lymphoma and multicentric Castleman’s disease [140]. Like other gammaherpesviruses, it is also a large (120–150 nm)) double-stranded DNA virus with more than 85 open reading frames (ORFs).

Genome sequencing of KSHV revealed that its ORF 4 encodes a protein that is structurally similar to the human RCA proteins [18]. As a result of alternative splicing, the ORF 4 protein is thought to be expressed as a membrane form (4 CCPs with a transmembrane domain) and a soluble form (only 4 CCP domain) [140].

The recombinant soluble form (4 CCPs) of the KSHV ORF 4 protein was found to block cell surface deposition of C3b, inhibit complement-mediated lysis of erythrocytes, and inactivate both C3b and C4b with the help of factor-I. Hence, the protein gained importance as a complement regulator and was named as Kaposica (KSHV inhibitor of complement activation) [19]. A parallel study also assigned similar functions to the protein and named KCP (KSHV complement control protein) [20]. Dissection of the functional domains of Kaposica revealed that CCPs 2–3 are essential for its ability to inactivate C3b and C4b, and CCPs 1–2 and CCPs 1–4 are required for the decay of the classical and alternative pathway C3 convertases, respectively [143]. Nevertheless, the study illustrated that all the CCPs are required for optimal activity. Spiller et al., [144] showed that Kaposica was expressed on KSHV infected cells as well as on the virion envelope, and possessed the potential to elude complement. Similar to human and viral regulators, the interaction of Kaposica with C3b and C4b depends on ionic interaction. The influence of electrostatic potential on the functional activities of Kaposica was studied by reducing or eliminating positive potential in the whole molecule, and by delineating the role of the electrostatic potential of the individual CCP modules [145]. These studies demonstrated the functional relevance to conservation of positive potential in CCPs 1 and 4 and the linkers of viral complement regulators. The molecular mechanism underlying the C3b/C4b inactivation process was unraveled by Gautam et al. [146]. It established that during C3b/C4b inactivation, Kaposica CCP2 provides a docking surface for factor I and CCP3 bridges the macroglobulin-2 and the CUB domains of C3b/C4b which stabilizes the CUB domain with respect to the core of the C3b/C4b. Further, the study revealed that such a mechanism is employed by human regulators too.

#### 3.2.10. Rhesus Rhadinovirus Complement Control Protein (RCP)

Rhesus monkey rhadinovirus (RRV) is a gammaherpesvirus which is genetically related to the KSHV. Its natural host is rhesus macaque but is also identified in squirrel monkeys, chimpanzees, gorillas, and pigtail macaques, to name a few. Because of its high genomic relatedness to KSHV, RRV endows an in vivo animal model of KSHV infection [147]. Interestingly, two independent strains of RRV have been identified: H26–95 and 17577. Genome sequencing and comparison showed that these strains are genetically similar. However, strain 17,577 is associated with disease in rhesus macaques following infection.

The primary sequence analysis of H26–95 and 17,577 strains revealed that ORF4 of both these isolates encode RCA-like proteins [148]. Using mass spectrometric analyses of RRV, O’Connor and Kedes [149] identified 33 virion-associated proteins which included seven envelope proteins. One among these envelope proteins was the RCA homolog which they referred to as rhesus rhadinovirus complement control protein (RCP). Later, molecular characterization of RCP of both the strains showed that they differ in the structure owing to the variation in their CCP content. The H26–95 isolate encoded a transcript with four CCP domains, whereas 17,577 isolate encoded a transcript having eight CCP domains. The RCP encoded by the former was referred to as RCP-H and that encoded by latter was referred to as RCP-1; both have a C-terminal transmembrane domain [41]. Expression of these proteins on the CHO cells followed by exposure of the cells to human serum resulted in inhibition of C3b deposition onto the cell surface, confirming their ability to inhibit complement [41]. The detailed analysis demonstrated that both the proteins are capable of binding to C3b and C4b and decaying the classical pathway C3 convertase. RCP-1 was also very efficient in decaying the alternative pathway C3 convertase, which was unusual for a viral RCA protein. Both RCP-H and RCP-1 also supported the inactivation of C3b and C4b with the assistance of factor I. As expected, RCP-1 was more potent than RCP-H [42].

#### 3.2.11. Murine Gammaherpesvirus 68 (γHV68) RCA Protein

Murine gammaherpesvirus 68 (γHV68/MHV68) is closely related to Epstein-Barr virus (EBV), HVS, and KSHV [150,151]. Originally isolated from a bank vole, it is a natural pathogen in several mice strains affecting several organs and causing acute infection in their peritoneal and spleen cells.

Complete sequence analysis of the γHV68 genome exhibited the presence of an open reading frame sharing homology with HVS and KSHV ORF-4 [151]. γHV68 RCA protein comprises four CCP domains and is expressed both as a membranous and a secretory form. It is suggested that the soluble form is generated by the proteolytic processing of the membranous form. Functional studies on this RCA homolog showed evidence that it effectively prevents the deposition of murine C3 on activating particles like zymosan by regulating the function of both the classical and alternative pathway C3 convertases, and is thereby a potent inhibitor of complement activation [43]. It has also been demonstrated that the mutant virus lacking γHV68 RCA protein had attenuated virulence as compared to the wild-type virus in complement sufficient mice, indicating that the γHV68 RCA protein plays a critical role in determining pathogenicity. Further, the mutant virus lacking the γHV68 RCA protein and wild-type γHV68 virus were equally virulent in C3-knockout mice, elucidating that γHV68 RCA targets host C3 [152]. The authors further demonstrated that while the protein could counter the effects of complement during acute infection efficiently, it was ineffective at subverting complement effects on viral latency [152]. It would be interesting to obtain further insights into its mode of action.

#### 3.2.12. Herpesvirus Saimiri CD59

In addition to a homolog of RCA proteins (discussed above), HVS also encodes a homolog of human CD59. In humans, CD59 functions as a potent inhibitor of complement-mediated cytolysis. It exerts this effect by tightly binding to C5b-8 and C5b-9 and inhibiting the incorporation of multiple C9 molecules into the membrane attack complex [153]. Additionally, CD59 is involved in T cell activation by serving as a ligand for CD2 [154].

Overall, the structure of HVS CD59 is similar to that of human CD59; however, specific differences are apparent. Like that of human CD59, it is also formed by 77 amino acids and contains a C-terminal glycosyl-phosphatidylinositol signal and attachment site. Moreover, the ten cysteines that form five disulfide bonds and stabilize the structure are also conserved in HVS CD59. The protein conversely lacks the N-linked glycosylation site and exhibits 48% amino acid identity to human CD59.

HVS CD59 has not been studied well like viral RCA proteins. Nevertheless, it was examined for its ability to block the MAC-mediated lysis of cells. In a transfection experiment, a comparison of its complement regulatory activity with human and squirrel monkey CD59 showed that it is as potent as these counterparts in inhibiting the human and monkey complement-mediated cytolysis, respectively [23,24]. Interestingly, it showed a broader species specificity compared to human CD59 in that it was also potent in inhibiting rat complement [23]. However, the structural basis for its broader specificity is not known. Efforts were also made to examine whether HVS CD59 is expressed on the virus-infected cells. Monkey cells, lytically infected with the HVS, showed the transcription of HVS CD59 gene [23]. It would be interesting to examine whether HVS CD59 can interact with CD2 and contribute to T cell activation, considering HVS is a T cell tropic virus.

## 4. Concluding Remarks

Successful infection of hosts by viruses is contingent upon their ability to subvert the vital arms of innate immunity. Consistent with this premise, viruses have been shown to employ both structural as well as non-structural proteins for evasion of complement. As described here, some of these proteins show homology to cellular genes. It is believed that such genes have been acquired from the host by horizontal gene transfer [155]. Viral proteins which do not show homology to the cellular proteins are unique to viruses but may harbor motif(s) which are similar to the cellular complement interacting proteins.

It is conceivable that generation of optimum innate and adaptive immune responses are essential for controlling viral infections and therefore, viruses subvert these to ensure their own survival within the host. However, it is also a fact that occasionally immune responses against virus-infected cells become overtly tissue damaging. Therefore, the pertinent question to be asked is, do some viruses encode complement regulators to reduce the complement-mediated tissue injury? This question has been addressed by two in vivo studies. Miller et al. [55] noticed that the cowpox virus deleted of IMP produced greater tissue damage accompanied by induration and hemorrhage compared to the wild-type virus. Similarly, Estep et al. [132] observed that monkeypox virus lacking MOPICE produced more severe disease in monkeys than the parental virus strain suggesting increased inflammation and tissue damage. Thus, viral complement regulators can serve as an entity that contributes to virulence by reducing immune responses [119,152] or as an entity that blocks the complement-mediated inflammation and thereby the tissue damage [55,132]. The latter paradigm is consistent with the views of Dubos who proposed that with time a state of peaceful coexistence is maintained between the host and parasite [156].

It is clear from the data discussed here that viruses encode diverse molecules to evade complement attack, which directly target complement proteins, enzymes, and complexes, and also inhibit the synthesis of complement proteins. Viruses, however, have additional means to counteract immune responses. For example, they are known to encode microRNAs to downregulate innate immune effectors’ expression and modulate the cellular microRNAs to enhance their replication [157]. Likewise, they also employ post-translational regulation of antiviral molecules by ubiquitin-dependent degradation [158]. Whether viruses exploit such mechanisms to elude complement attack is unknown, and a careful look into these may enrich this field with more diverse mechanisms.

Up until recently, the complement system was defined as a humoral system. However, the system has been shown to operate intracellularly affecting T cell effector function [159]. Further, it has been shown that intracellular sensing of C3b opsonized viruses activate mitochondrial antiviral signaling (MAVS)-dependent signaling cascade leading to secretion of proinflammatory cytokines and proteasomal degradation of viral particles [9]. It, therefore, is likely that viruses may have devised a mechanism that restricts complement opsonization inside the cells, which requires further investigation.

Understanding of the intracellular complement system, dubbed the complosome, is still limited. We now know that inside the cell, complement system crosstalks with MAVS [9] and inflammasome [160], but it is unwise to conclude that the crosstalks are limited only with the aforementioned cellular pathways. A precise understanding of the interaction of the complosome with different antiviral molecular processes and how viral complement regulators antagonize these would provide a larger picture of how viruses have adapted to the complement threat.

## Figures and Tables

**Figure 1 viruses-13-00208-f001:**
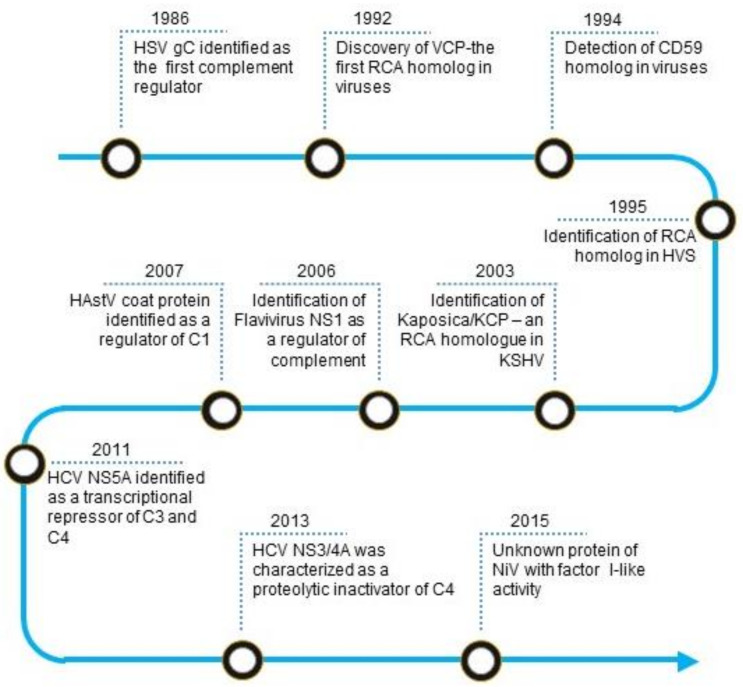
Timeline of identification of virally encoded complement regulators. Abbreviations: HSV gC, Herpes simplex virus glycoprotein C; VCP, Vaccinia virus complement control protein; RCA, regulator of complement activation; HVS, Herpesvirus saimiri; KSHV, Kaposi’s sarcoma-associated herpesvirus; Kaposica, KSHV inhibitor of complement activation; KCP, KSHV complement control protein; NS1, Non-structural protein; HAstV coat protein, Human astrovirus coat protein; HCV NS5A, Hepatitis C virus non-structural 5A protein; HCV NS3/4A, Hepatitis C virus non-structural 3/4A protease; NiV, Nipah virus.

**Figure 2 viruses-13-00208-f002:**
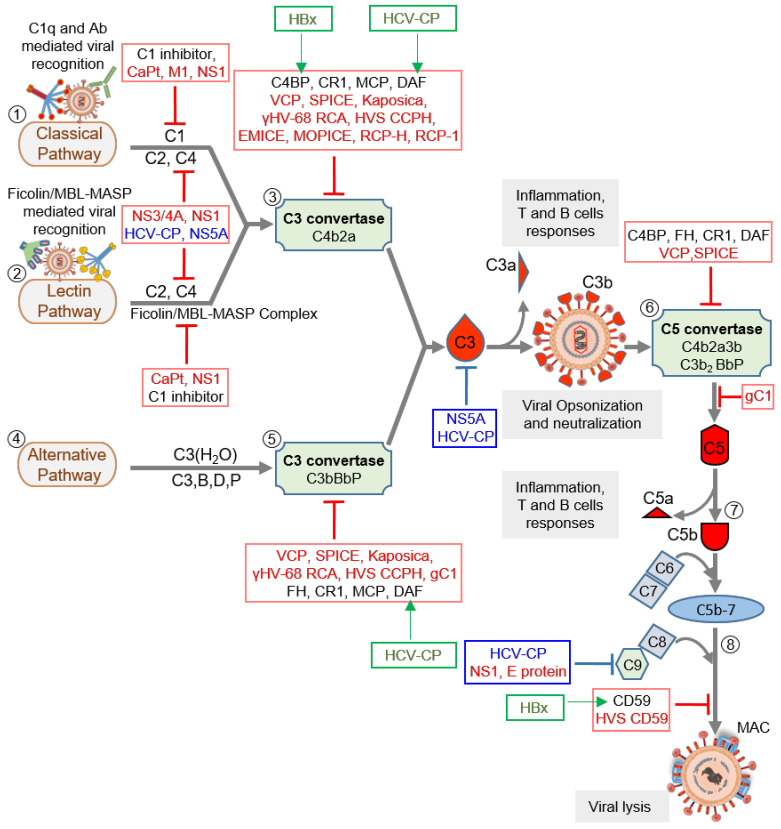
Complement activation and its regulation by host regulators and virally encoded molecules. Viruses can activate the host complement system by three major pathways: classical pathway (CP), lectin pathway (LP), and alternative pathway (AP). (1) In CP activation, viruses are known to be recognized by C1q and antibody. (2) In LP activation, viruses are recognized by the Ficolin/MBL-MASP complex. (3) Recognition is followed by activation of C1 and Ficolin/MBL complexes, in CP and LP respectively, which results in the cleavage of C4 and C2 and formation of C3 convertase C4b2a. The convertase then cleaves C3 into C3a and C3b. The latter opsonizes viral particles. (4) In AP, viruses are recognized directly by C3b molecules, which are generated by the initial C3 convertase C3(H2O)Bb. (5) The surface-bound C3b molecules then trigger the formation of C3 convertase C3bBb with the help of factors B and D, which is stabilized by properdin (P). The C3 convertases formed then cleaves more C3 to opsonize viral particles. (6) When C3b is attached to the preformed C3 convertase (C4b2a or C3bBbP), it is converted into C5 convertase (C4b2a3b or C3b2BbP) which is capable of cleaving C5 into C5a and C5b. (7) The newly formed C5b combines C6 and C7 to form a C5b-7 trimer that can attach to the viral surface. (8) Binding of the trimer to C8 and C9 followed by polymerization of C9 results in the formation of the membrane attack complex (MAC) that induces virolysis. These pathways are regulated at various steps by host complement regulators like C1 inhibitor, C4b-binding protein (C4BP), complement receptor 1 (CR1; CD35), membrane cofactor protein (MCP; CD46), decay-accelerating factor (DAF; CD55), factor H (FH) and CD59. Viral complement regulators that target complement proteins, enzymes, and complexes are shown in red text, whereas those that inhibit complement proteins’ expression are shown in blue text. Some viral complement regulators enhance the expression of host complement regulators. These are identified in green text, and green arrows mark the regulator they upregulate. Abbreviations: CaPt, Astrovirus capsid protein; M1, Influenza virus matrix protein 1; NS1, Flavivirus non-structural protein 1; NS3/4A, Hepatitis C virus non-structural 3/4A protease; HCV-CP, Hepatitis C virus core protein; NS5A, Hepatitis C virus non-structural 5A protein; VCP, Vaccinia virus complement control protein; SPICE, Smallpox inhibitor of complement enzymes; Kaposica, KSHV inhibitor of complement activation; γHV-68 RCA, Murine gammaherpesvirus 68 regulator of complement activation; HVS CCPH, Herpesvirus saimiri complement control protein homolog; EMICE, Ectromelia virus inhibitor of complement enzymes; MOPICE, Monkeypox inhibitor of complement enzymes; RCP-H, Rhesus rhadinovirus complement control protein H; RCP-1, Rhesus rhadinovirus complement control protein-1; gC1, Herpes simplex virus glycoprotein C-1; E protein, Zika virus E protein, HVS CD59, Herpesvirus saimiri CD59; HBx, Hepatitis B virus X protein.

**Table 1 viruses-13-00208-t001:** Viruses and their complement regulators.

Virus Family	Virus	Complement Evasion Protein	Target	Reference
*Herpesviridae*	Herpes simplex virus	Glycoprotein C-1	AP C3 convertase and C3b	[13,38]
	Herpesvirus saimiri	CD59 homolog	C5b-8 and C5b-9	[23,24]
		HVS CCPH	CP/LP and AP C3 convertase	[39,40]
	Kaposi sarcoma-associated herpesvirus	Kaposica/KCP	CP/LP and AP C3 convertase	[19,20]
	Rhesus rhadinovirus	RCP-H and RCP-1	CP/LP and AP C3 convertase	[41,42]
	Murine gammaherpesvirus 68	γHV68 RCA protein	CP/LP and AP C3 convertase	[43]
*Astroviridae*	Astrovirus	CaPt	C1q and MBL	[31,44]
*Orthomyxoviridae*	Influenza virus	M1	C1q	[32]
*Paramyxoviridae*	Nipah virus	Unknown protein with factor I like activity	C3b	[33]
*Togaviridae*	Chikungunya virus	Unknown protein with factor I like activity	C3b	[45]
*Flaviviridae*	Hepatitis C virus	Core protein	C3, C4 and C9 genes	[28,29,30]
		NS3/4A	C4	[46]
		NS5A	C3 and C4 genes	[28,47]
	West Nile virus and Dengue virus	NS1	C4 and C9	[26,27]
	Zika virus	E protein	C5b-6, C8 and C9	[48]
*Hepadnaviridae*	Hepatitis B virus	HBx protein	CD59 and C4BP genes	[49,50]
*Poxviridae*	Variola virus	SPICE	CP/LP and AP C3/C5 convertase	[51,52]
	Vaccinia virus	VCP	CP/LP and AP C3/C5 convertase	[17,53,54]
	Cowpox virus	IMP	Unknown	[55]
	Monkeypox virus	MOPICE	CP/LP C3 convertase	[56]
	Ectromelia virus	EMICE	CP/LP C3 convertase	[57]

CP—Classical pathway, AP—Alternative pathway, LP—Lectin pathway.

## Data Availability

Not applicable.

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
