# Peer review of "Virus-Encoded Complement Regulators: Current Status"

_viruses, 2021, doi:10.3390/v13020208_

Round 1

Reviewer 1 Report

This is a well written and very comprehensive review of strategies used by viruses to overcome the complement system. It provides a broad coverage of the history of research into the field and provides a detailed and helpful list of references. It is certainly worthy of publication.

There are a few minor comments/corrections that might improve the manuscript:

  1. Line 23. Remove “…and are the most…” and substitute “…to become…”
  2. Line 29. Add “humoral” to immunity (i.e., “…Ehrlich established the idea of humoral immunity.”)
  3. Line 53. Remove “all”.
  4. Figure 1 timeline. Nice figure. Recommend you describe the abbreviations in the figure legend (e.g., HSV gC, VCP, RCA, HVS, HastV etc). An alternate suggestion is to provide a sidebar that lists abbreviations utilized in both Figs (Figs 1/2).
  5. Line 102: reference problem, #24 listed twice (also, it is difficult to see the separated references; would consider commas to separate references in the entire manuscript).
  6. Figure 2. Nice figure also. Would help to describe other abbreviations such as CaPt, Hbx, etc. Perhaps as suggested in point 1, might be better to just provide a separate box for the the numerous abbreviations used.
  7. Lines 386/387. Please rewrite and explain sentence better: “Notably, the former was used as a vaccine to eradicate the latter.” Are you saying vaccinia causes outbreaks in dairy cattle but was used as a vaccine in dairy cattle? Or do you mean to say, vaccinia was used as a vaccine against cowpox?
  8. Line 424: correct spelling of author’s name to “Liszewski.”
  9. Line 426. Consider adding after the last sentence: “The same group determined that SPICE attaches to cells via its glycosaminoglycans to efficiently regulate complement on the cell surface.” Ref: https://pubmed.ncbi.nlm.nih.gov/18768877/
  10. Line 484-485. This concept is rather confusing. To be consistent with your statements in Concluding remarks, consider revising by saying “…suggesting the protein contributes to virulence by reducing the immune response (see Concluding remarks).”
  11. Line 573. Consider deleting ‘the latter’ and substituting “strain 17577” to clarify.
  12. Line 635. Remove the ‘s’ after “…contributes to T cells…”. Should be “T cell.”
  13. Line 638. Delete “the” before “innate immunity.”
  14. Optional idea: It would be helpful to have a table that summarizes the described viruses and their complement regulators.

Author Response

Reviewer #1

The reviewer liked the review and mentioned that it is a very well written and comprehensive review. We sincerely thank the reviewer for this and the comments (below).

  1. Line 23. Remove “…and are the most…” and substitute “…to become…”

Response: “…and are the most…” has been edited to “to become”.

  1. Line 29. Add “humoral” to immunity (i.e., “…Ehrlich established the idea of humoral immunity.”)

Response:  “humoral” has been incorporated.

  1. Line 53. Remove “all”.

Response: “all” has been removed from the text.

  1. Figure 1 timeline. Nice figure. Recommend you describe the abbreviations in the figure legend (e.g., HSV gC, VCP, RCA, HVS, HastV etc). An alternate suggestion is to provide a sidebar that lists abbreviations utilized in both Figs (Figs 1/2).

Response: We have incorporated the abbreviations in the legend.

  1. Line 102: reference problem, #24 listed twice (also, it is difficult to see the separated references; would consider commas to separate references in the entire manuscript).

Response: Extra reference #24 has been deleted, and formatting has been revised, which adds commas to separate references.

  1. Figure 2. Nice figure also. Would help to describe other abbreviations such as CaPt, Hbx, etc. Perhaps as suggested in point 1, might be better to just provide a separate box for the numerous abbreviations used.

Response: As suggested, we have now added the abbreviations in the figure legend.

  1. Lines 386/387. Please rewrite and explain sentence better: “Notably, the former was used as a vaccine to eradicate the latter.” Are you saying vaccinia causes outbreaks in dairy cattle but was used as a vaccine in dairy cattle? Or do you mean to say, vaccinia was used as a vaccine against cowpox?

Response: The sentence has been revised to provide better clarity.

  1. Line 424: correct spelling of author’s name to “Liszewski.”

Response: We are sorry for misspelling the name. It has been corrected.

  1. Line 426. Consider adding after the last sentence: “The same group determined that SPICE attaches to cells via its glycosaminoglycans to efficiently regulate complement on the cell surface.” Ref: https://pubmed.ncbi.nlm.nih.gov/18768877/

Response: The suggested change has been incorporated.

  1. Line 484-485. This concept is rather confusing. To be consistent with your statements in Concluding remarks, consider revising by saying “…suggesting the protein contributes to virulence by reducing the immune response (see Concluding remarks).”

Response: We have deleted the line to remove confusion.

  1. Line 573. Consider deleting ‘the latter’ and substituting “strain 17577” to clarify.

Response: “the latter” has been edited to “strain 17577”

  1. Line 635. Remove the ‘s’ after “…contributes to T cells…”. Should be “T cell.”

Response: Edited as suggested.

  1. Line 638. Delete “the” before “innate immunity.”

Response: Edited.

  1. Optional idea: It would be helpful to have a table that summarizes the described viruses and their complement regulators.

Response: We have now added a Table.

Reviewer 2 Report

A comprehensive and clear written manuscript which covers most of recent and historic findings. 

Minor Points:

Please give references to your Statements between line 52 to 61.

line 122 skip "the" in "…. must target both the phases of the Virus life cycle"  

In chapter 3.2.2 the role of NS1 is described in detail. Please include the recent finding that the E Protein of ZikV nac also interfere with the fomation of the MAC (Malekshahi Z et al Front. Immunol 2020 Oct 28;11:569549. doi: 10.3389/fimmu.2020.569549).   

Author Response

We thank the reviewer for appreciating our efforts. The comments are addressed below.

  1. Please give references to your Statements between line 52 to 61.

Response: We have now added a reference where required.

  1. line 122 skip "the" in "…. must target both the phases of the Virus life cycle"

Response: “the” has been removed and the sentence.

  1. In chapter 3.2.2 the role of NS1 is described in detail. Please include the recent finding that the E Protein of ZikVnac also interferes with the formation of the MAC (Malekshahi Z et al Front. Immunol 2020 Oct 28;11:569549. doi: 10.3389/fimmu.2020.569549).

Response: A separate section elucidating Zika virus E protein's role in evading complement has been added.